# GENERATIVE RESTRICTED KERNEL MACHINES

## ABSTRACT

We introduce a novel framework for generative models based on Restricted Kernel Machines (RKMs) with multi-view generation and uncorrelated feature learning capabilities, called Gen-RKM. To incorporate multi-view generation, this mechanism uses a shared representation of data from various views. The mechanism is flexible to incorporate both kernel-based, (deep) neural network and Convolutional based models within the same setting. To update the parameters of the network, we propose a novel training procedure which jointly learns the features and shared subspace representation. The latent variables are given by the eigendecomposition of the kernel matrix, where the mutual orthogonality of eigenvectors represents uncorrelated features. Experiments demonstrate the potential of the framework through qualitative and quantitative evaluation of generated samples on various standard datasets.

## 1 INTRODUCTION

In the past decade, interest in generative models has grown tremendously, finding applications in multiple fields such as, generated art, on-demand video, image denoising (Vincent et al., 2010), exploration in reinforcement learning (Florensa et al., 2018), collaborative filtering (Salakhutdinov et al., 2007), inpainting (Yeh et al., 2017) and many more.

Some examples of graphical models based on a probabilistic framework with latent variables are Variational Auto-Encoders (Kingma & Welling, 2014) and Restricted Boltzmann Machines (RBMs) (Smolensky, 1986; Salakhutdinov & Hinton, 2009). More recently proposed models are based on adversarial training such as Generative Adversarial Networks (GANs) (Goodfellow et al., 2014) and its many variants. Furthermore, auto-regressive models such as Pixel Recurrent Neural Networks (PixelRNNs) (Van Den Oord et al., 2016) model the conditional distribution of every individual pixel given previous pixels. All these approaches have their own advantages and disadvantages. For example, RBMs perform both learning and Bayesian inference in graphical models with latent variables. However, such probabilistic models must be properly normalized, which requires evaluating intractable integrals over the space of all possible variable configurations (Salakhutdinov & Hinton, 2009). Currently GANs are considered as the state-of-the-art for generative modeling tasks, producing high-quality images but are more difficult to train due to unstable training dynamics, unless more sophisticated variants are applied.

Many datasets are comprised of different representations of the data, or views. Views can correspond to different modalities such as sounds, images, videos, sequences of previous frames, etc. Although each view could individually be used for learning tasks, exploiting information from all views together could improve the learning quality (Pu et al., 2016; Liu & Tuzel, 2016; Chen & Denoyer, 2017). Also, it is among the goals of the latent variable modelling to model the description of data in terms of *uncorrelated or independent* components. Some classical examples are Independent Component Analysis; Hidden Markov models (Rabiner & Juang, 1986); Probabilistic Principal Component Analysis (PCA) (Tipping & Bishop, 1999); Gaussian-Process Latent variable model (Lawrence, 2005) and factor analysis. Hence, when learning a latent space in generative models, it becomes interesting to find a disentangled representation. Disentangled variables are generally considered to contain interpretable information and reflect separate factors of variation in the data for e.g. lighting conditions, style, colors, etc. The definition of disentanglement in the literature is not precise, however many believe that a representation with statistically independent variables is a good starting point (Schmidhuber, 1992; Ridgeway, 2016). Such representations extract information into a compact form which makes it possible to generate samples with specific characteristics

(Chen et al., 2018; Bouchacourt et al., 2018; Tran et al., 2017; Chen et al., 2016). Additionally, these representations have been found to generalize better and be more robust against adversarial attacks (Alemi et al., 2017).

In this work, we propose an alternative generative mechanism based on the framework of Restricted Kernel Machines (RKMs) (Suykens, 2017), called Generative RKM (Gen-RKM). RKMs yield a representation of kernel methods with visible and hidden units establishing links between Kernel PCA, Least-Squares Support Vector Machines (LS-SVM) (Suykens et al., 2002) and RBMs. This framework has a similar energy form as RBMs, though there is a non-probabilistic training procedure where the eigenvalue decomposition plays the role of normalization. Recently, Houthuys & Suykens (2018) used this framework to develop tensor-based multi-view classification models and Schreurs & Suykens (2018) showed how kernel PCA fits into this framework.

**Contributions. 1)** A novel multi-view generative model based on the RKM framework where multiple views of the data can be generated simultaneously. **2)** Two methods are proposed for computing the pre-image of the feature vectors: with the feature map explicitly known or unknown. We show that the mechanism is flexible to incorporate both kernel-based, (deep) convolutional neural network based models within the same setting. **3)** When using explicit feature maps, we propose a training algorithm that jointly performs the feature-selection and learns the common-subspace representation in the same procedure. **4)** Qualitative and quantitative experiments demonstrate that the model is capable of generating good quality images of natural objects. Further experiments on multi-view datasets exhibit the potential of the model. Thanks to the orthogonality of eigenvectors of the kernel matrix, the learned latent variables are uncorrelated. This resembles a disentangled representation, which makes it possible to generate data with specific characteristics.

This paper is organized as follows. In Section 2, we discuss the Gen-RKM training and generation mechanism when multiple data sources are available. In Section 3, we explain how the model incorporates both kernel methods and neural networks through the use of implicit and explicit feature maps respectively. When the feature maps are defined by neural networks, the Gen-RKM algorithm is explained in Section 4. In Section 5, we show experimental results of our model applied on various public datasets. Section 6 concludes the paper along with directions towards the future work. Additional supplementary materials are given in the Appendix A.

## 2    GENERATIVE RESTRICTED KERNEL MACHINES FRAMEWORK

The proposed Gen-RKM framework consists of two phases: a training phase and a generation phase which occurs one after another.

### 2.1    TRAINING

Similar to Energy-Based Models (EBMs, see LeCun et al. (2004) for details), the RKM objective function captures dependencies between variables by associating a scalar energy to each configuration of the variables. Learning consists of finding an energy function in which the observed configurations of the variables are given lower energies than unobserved ones. Note that the schematic representation, as shown in Figure 1 is similar to Discriminative RBMs (Larochelle & Bengio, 2008) and the objective function $\mathcal{J}_t$ (defined below) has an energy form similar to RBMs with additional regularization terms. The latent space dimension in the RKM setting has a similar interpretation as the number of hidden units in a restricted Boltzmann machine, where in the specific case of the RKM these hidden units are uncorrelated.

We assume a dataset $\mathcal{D} = \{\boldsymbol{x}_i, \boldsymbol{y}_i\}_{i=1}^{N}$, with $\boldsymbol{x}_i \in \mathbb{R}^d$, $\boldsymbol{y}_i \in \mathbb{R}^p$ comprising of $N$ data points. Here $\boldsymbol{y}_i$ may represent an additional view of $\boldsymbol{x}_i$, e.g., an additional image from a different angle, the caption of an image or a class label. Starting from the RKM interpretation of Kernel PCA, which gives an upper bound on the equality constrained Least-Squares Kernel PCA objective function (Suykens, 2017), and applying the feature-maps $\phi_1 : \mathbb{R}^d \mapsto \mathbb{R}^{d_f}$ and $\phi_2 : \mathbb{R}^p \mapsto \mathbb{R}^{p_f}$ to the input data points,

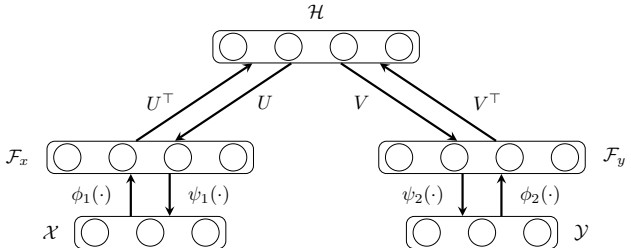

Figure 1: Gen-RKM schematic representation modeling a common subspace $\mathcal{H}$ between two data sources $\mathcal{X}$ and $\mathcal{Y}$. The $\phi_1$, $\phi_2$ are the feature maps ($\mathcal{F}_x$ and $\mathcal{F}_y$ represent the feature-spaces) corresponding to the two data sources. While $\psi_1$, $\psi_2$ represent the pre-image maps. The interconnection matrices $\boldsymbol{U}, \boldsymbol{V}$ model dependencies between latent variables and the mapped data sources.

the training objective function $\mathcal{J}_t$ for generative RKM is given by[1]:

$$\mathcal{J}_t = \sum_{i=1}^{N} \left( -\phi_1(\boldsymbol{x}_i)^\top \boldsymbol{U} \boldsymbol{h}_i - \phi_2(\boldsymbol{y}_i)^\top \boldsymbol{V} \boldsymbol{h}_i + \frac{\lambda}{2} \boldsymbol{h}_i^\top \boldsymbol{h}_i \right) + \frac{\eta_1}{2} \operatorname{Tr}(\boldsymbol{U}^\top \boldsymbol{U}) + \frac{\eta_2}{2} \operatorname{Tr}(\boldsymbol{V}^\top \boldsymbol{V}), \quad (1)$$

where $\boldsymbol{U} \in \mathbb{R}^{d_f \times s}$ and $\boldsymbol{V} \in \mathbb{R}^{p_f \times s}$ are the unknown interaction matrices, and $\boldsymbol{h}_i \in \mathbb{R}^s$ are the latent variables modeling a common subspace $\mathcal{H}$ between the two input spaces $\mathcal{X}$ and $\mathcal{Y}$ (see Figure 1). The derivation of this objective function is given in the Appendix A.1.

Given $\eta_1 > 0$ and $\eta_2 > 0$ as regularization parameters, the stationary points of $\mathcal{J}_t$ are given by:

$$\begin{cases} \frac{\partial \mathcal{J}_t}{\partial \boldsymbol{h}_i} = 0 \implies \lambda \boldsymbol{h}_i = \boldsymbol{U}^\top \phi_1(\boldsymbol{x}_i) + \boldsymbol{V}^\top \phi_2(\boldsymbol{y}_i), \ \ \forall i = 1, \ldots, N \\ \frac{\partial \mathcal{J}_t}{\partial \boldsymbol{U}} = 0 \implies \boldsymbol{U} = \frac{1}{\eta_1} \sum_{i=1}^{N} \phi_1(\boldsymbol{x}_i) \boldsymbol{h}_i^\top \\ \frac{\partial \mathcal{J}_t}{\partial \boldsymbol{V}} = 0 \implies \boldsymbol{V} = \frac{1}{\eta_2} \sum_{i=1}^{N} \phi_2(\boldsymbol{y}_i) \boldsymbol{h}_i^\top. \end{cases} \quad (2)$$

Substituting $\boldsymbol{U}$ and $\boldsymbol{V}$ in the first equation above, denoting $\boldsymbol{\Lambda} = \operatorname{diag}\{\lambda_1, \ldots, \lambda_s\} \in \mathbb{R}^{s \times s}$ with $s \leq N$, yields the following eigenvalue problem:

$$\left[ \frac{1}{\eta_1} \boldsymbol{K}_1 + \frac{1}{\eta_2} \boldsymbol{K}_2 \right] \boldsymbol{H}^\top = \boldsymbol{H}^\top \boldsymbol{\Lambda}, \quad (3)$$

where $\boldsymbol{H} = \left[ \boldsymbol{h}_1, \ldots, \boldsymbol{h}_N \right] \in \mathbb{R}^{s \times N}$ with $s \leq N$ is the number of selected principal components and $\boldsymbol{K}_1, \boldsymbol{K}_2 \in \mathbb{R}^{N \times N}$ are the kernel matrices corresponding to data sources[2]. Based on Mercer's theorem (Mercer, 1909), positive-definite kernel functions $k_1 : \mathbb{R}^d \times \mathbb{R}^d \mapsto \mathbb{R}$, $k_2 : \mathbb{R}^p \times \mathbb{R}^p \mapsto \mathbb{R}$ can be defined such that $k_1(\boldsymbol{x}_i, \boldsymbol{x}_j) = \langle \phi_1(\boldsymbol{x}_i), \phi_1(\boldsymbol{x}_j) \rangle$, and $k_2(\boldsymbol{y}_i, \boldsymbol{y}_j) = \langle \phi_2(\boldsymbol{y}_i), \phi_2(\boldsymbol{y}_j) \rangle$, $\forall i, j = 1, \ldots, N$ forms the elements of corresponding kernel matrices. The feature maps $\phi_1$ and $\phi_2$, mapping the input data to the high-dimensional feature space (possibly infinite) are implicitly defined by kernel functions. Typical examples of such kernels are given by the Gaussian RBF kernel $k(\boldsymbol{x}_i, \boldsymbol{x}_j) = e^{-\|\boldsymbol{x}_i - \boldsymbol{x}_j\|_2^2 / (2\sigma^2)}$ or the Laplace kernel $k(\boldsymbol{x}_i, \boldsymbol{x}_j) = e^{-\|\boldsymbol{x}_i - \boldsymbol{x}_j\|_2 / \sigma}$ just to name a few (Scholkopf & Smola, 2001). However, one can also define explicit feature maps, still preserving the positive-definiteness of the kernel function by construction (Suykens et al., 2002).

## 2.2 GENERATION

In this section, we derive the equations for the generative mechanism. RKMs resembling energy-based models, the inference consists in clamping the value of observed variables and finding configurations of the remaining variables that minimizes the energy (LeCun et al., 2004). Given the

---

[1]For convenience, it is assumed that all the feature vectors are centered in the feature space $\mathcal{F}$ using $\tilde{\phi}(\boldsymbol{x}) := \phi(\boldsymbol{x}) - \frac{1}{N} \sum_{i=1}^{N} \phi(\boldsymbol{x}_i)$. Otherwise, a centered kernel matrix could be obtained using Eq. 17 (Appendix A.4).
[2]While in the above section we have assumed that only two data sources (namely $\mathcal{X}$ and $\mathcal{Y}$) are available for learning, the above procedure could be extended to multiple data-sources. For the $M$ views or data-sources, this yields the training problem: $\left[ \sum_{\ell=1}^{M} \frac{1}{\eta_\ell} \boldsymbol{K}_\ell \right] \boldsymbol{H}^\top = \boldsymbol{H}^\top \boldsymbol{\Lambda}$.

learned interconnection matrices $U$ and $V$, and a given latent variable $h^\star$, consider the following objective function:

$$\mathcal{J}_g = -\phi_1(x^\star)^\top U h^\star - \phi_2(y^\star)^\top V h^* + \frac{1}{2}\phi_1(x^\star)^\top \phi_1(x^\star) + \frac{1}{2}\phi_2(y^\star)^\top \phi_2(y^\star), \qquad (4)$$

with an additional regularization term on data sources. Here $\mathcal{J}_g$ denotes the objective function for generation. The given latent variable $h^\star$ can be the corresponding latent code of a training point, a newly sampled hidden unit or a specifically determined one. Above cases correspond to generating the reconstructed visible unit, generating a random new visible unit or exploring the latent space by carefully selecting hidden units respectively. The stationary points of $\mathcal{J}_g$ are characterized by:

$$\begin{cases} \frac{\partial \mathcal{J}_g}{\partial \phi_1(x^\star)} = 0 \implies \phi_1(x^\star) = U h^\star, \\ \frac{\partial \mathcal{J}_g}{\partial \phi_2(y^\star)} = 0 \implies \phi_2(y^\star) = V h^\star. \end{cases} \qquad (5)$$

Using $U$ and $V$ from Eq. 2, we obtain the generated feature vectors:

$$\phi_1(x^\star) = \left( \frac{1}{\eta_1} \sum_{i=1}^{N} \phi_1(x_i) h_i^\top \right) h^\star, \quad \phi_2(y^\star) = \left( \frac{1}{\eta_2} \sum_{i=1}^{N} \phi_2(y_i) h_i^\top \right) h^\star. \qquad (6)$$

To obtain the generated data, one now needs to compute the inverse images of the feature maps $\phi_1(\cdot)$ and $\phi_2(\cdot)$ in the respective input spaces, i.e., solve the *pre-image problem*. We seek to find the functions $\psi_1 \colon \mathbb{R}^{d_f} \mapsto \mathbb{R}^d$ and $\psi_2 \colon \mathbb{R}^{p_f} \mapsto \mathbb{R}^p$ corresponding to the two data-sources, such that $(\psi_1 \circ \phi_1)(x^\star) \approx x^\star$ and $(\psi_2 \circ \phi_2)(y^\star) \approx y^\star$, where $\phi_1(x^\star)$ and $\phi_2(y^\star)$ are calculated using Eq. 6.

When using kernel methods, explicit feature maps are not necessarily known. Commonly used kernels such as the radial-basis function and polynomial kernels map the input data to a very high dimensional feature space. Hence finding the pre-image, in general, is known to be an ill-conditioned problem (Mika et al., 1999). However, various approximation techniques have been proposed (Bui et al., 2019; Kwok & Tsang, 2003; Honeine & Richard, 2011; Weston et al., 2004) which could be used to obtain the approximate pre-image $\hat{x}$ of $\phi_1(x^\star)$. In section 3.1, we employ one such technique to demonstrate the applicability in our model, and consequently generate the multi-view data. One could also define explicit pre-image maps. In section 3.2, we define parametric pre-image maps and learn the parameters by minimizing the appropriately defined objective function. The next section describes the above two pre-image methods for both cases, i.e., when the feature map is explicitly known or unknown, in greater detail.

## 3  IMPLICIT & EXPLICIT FEATURE MAP

### 3.1  IMPLICIT FEATURE MAP

As noted in the previous section, since $x^\star$ may not exist, we find an approximation $\hat{x}$. A possible technique is shown by Schreurs & Suykens (2018). Left multiplying Eq. 6 by $\phi_1(x_i)^\top$ and $\phi_2(y_i)^\top$, $\forall i = 1, \ldots, N$, we obtain:

$$k_{x^\star} = \frac{1}{\eta_1} K_1 H^\top h^\star, \quad k_{y^\star} = \frac{1}{\eta_2} K_2 H^\top h^\star, \qquad (7)$$

where, $k_{x^\star} = [k(x_1, x^\star), \ldots, k(x_N, x^\star)]^\top$ represents the *similarities* between $\phi_1(x^\star)$ and training data points in the feature space, and $K_1 \in \mathbb{R}^{N \times N}$ represents the centered kernel matrix of $\mathcal{X}$. Similar conventions follow for $\mathcal{Y}$ respectively. Using the *kernel-smoother* method (Hastie et al., 2001), the pre-images are given by:

$$\hat{x} = \psi_1(\phi_1(x^\star)) = \frac{\sum_{j=1}^{n_r} \tilde{k}_1(x_j, x^\star) x_j}{\sum_{j=1}^{n_r} \tilde{k}_1(x_j, x^\star)}, \quad \hat{y} = \psi_2(\phi_2(y^\star)) = \frac{\sum_{j=1}^{n_r} \tilde{k}_2(y_j, y^\star) y_j}{\sum_{j=1}^{n_r} \tilde{k}_2(y_j, y^\star)}, \qquad (8)$$

where $\tilde{k}_1(x_i, x^\star)$ and $\tilde{k}_2(y_i, y^\star)$ are the scaled similarities (see Eq. 8) between 0 and 1 and $n_r$ the number of closest points based on the similarity defined by kernels $\tilde{k}_1$ and $\tilde{k}_2$.

## 3.2 Explicit Feature map

While using an explicit feature map, Mercer's theorem is still applicable due to the positive semi-definiteness of the kernel function by construction, thereby allowing the derivation of Eq. 3. In the experiments, we use a set of (convolutional) neural networks as the feature maps $\phi_{\boldsymbol{\theta}}(\cdot)$. Another (transposed convolutional) neural network is used for the pre-image map $\psi_{\boldsymbol{\zeta}}(\cdot)$ (Dumoulin & Visin, 2016). The network parameters $\{\boldsymbol{\theta}, \boldsymbol{\zeta}\}$ are learned by minimizing the reconstruction errors defined by $\mathcal{L}_1(\boldsymbol{x}, \psi_{1_{\boldsymbol{\zeta}_1}}(\phi_{1_{\boldsymbol{\theta}_1}}(\boldsymbol{x})))$ and $\mathcal{L}_2(\boldsymbol{y}, \psi_{2_{\boldsymbol{\zeta}_2}}(\phi_{2_{\boldsymbol{\theta}_2}}(\boldsymbol{y})))$. In our experiments, we use the mean-squared errors $\mathcal{L}_1(\boldsymbol{x}, \psi_{1_{\boldsymbol{\zeta}_1}}(\phi_{1_{\boldsymbol{\theta}_1}}(\boldsymbol{x}))) = \frac{1}{N}\sum_{i=1}^{N} \left\| \boldsymbol{x}_i - \psi_{1_{\boldsymbol{\zeta}_1}}(\phi_{1_{\boldsymbol{\theta}_1}}(\boldsymbol{x}_i)) \right\|_2^2$ and $\mathcal{L}_2(\boldsymbol{y}, \psi_{2_{\boldsymbol{\zeta}_2}}(\phi_{2_{\boldsymbol{\theta}_2}}(\boldsymbol{y}))) = \frac{1}{N}\sum_{i=1}^{N} \left\| \boldsymbol{y}_i - \psi_{2_{\boldsymbol{\zeta}_2}}(\phi_{2_{\boldsymbol{\theta}_2}}(\boldsymbol{y}_i)) \right\|_2^2$, however, in principle, one can use any other loss appropriate to the dataset. Here $\phi_{1_{\boldsymbol{\theta}_1}}(\boldsymbol{x}_i)$ and $\phi_{2_{\boldsymbol{\theta}_2}}(\boldsymbol{y}_i)$ are computed from Eq. 6, i.e., the generated points in feature space from the subspace $\mathcal{H}$.

Adding the loss function directly into the objective function $\mathcal{J}_t$ is not suitable for minimization. Instead, we use the stabilized objective function defined as $\mathcal{J}_{stab} = \mathcal{J}_t + \frac{c_{\text{stab}}}{2}\mathcal{J}_t^2$, where $c_{stab} \in \mathbb{R}^+$ is the regularization constant (Suykens, 2017). This tends to push the objective function $\mathcal{J}_t$ towards zero, which is also the case when substituting the solutions $\lambda_i, \boldsymbol{h}_i$ back into $\mathcal{J}_t$ (see Appendix A.3 for details). The combined training objective is given by:

$$\min_{\boldsymbol{\theta}_1, \boldsymbol{\theta}_2, \boldsymbol{\zeta}_1, \boldsymbol{\zeta}_2} \mathcal{J}_c = \mathcal{J}_{stab} + \frac{c_{\text{acc}}}{2N}\left( \sum_{i=1}^{N} \left[ \mathcal{L}_1(\boldsymbol{x}_i, \psi_{1_{\boldsymbol{\zeta}_1}}(\phi_{1_{\boldsymbol{\theta}_1}}(\boldsymbol{x}_i))) + \mathcal{L}_2(\boldsymbol{y}_i, \psi_{2_{\boldsymbol{\zeta}_2}}(\phi_{2_{\boldsymbol{\theta}_2}}(\boldsymbol{y}_i))) \right] \right), \quad (9)$$

where $c_{\text{acc}} \in \mathbb{R}^+$ is a regularization constant to control the stability with reconstruction accuracy. In this way, we combine feature-selection and subspace learning within the same training procedure.

There is also an intuitive connection between Gen-RKM and autoencoders. Namely, the properties of kernel PCA resemble the objectives of the 3 variations of an autoencoder: standard (Kramer, 1991), VAE (Kingma & Welling, 2014) and $\beta$-VAE (Higgins et al., 2017). **1)** Similar to an autoencoder, Gen-RKM minimizes the reconstruction error in the loss function (see Eq. 9), where kernel PCA which acts as a denoiser (the information is compressed in the principal components). **2)** By interpreting kernel PCA within the LS-SVM setting (Suykens et al., 2002), the PCA analysis can take the interpretation of a one-class modeling problem with zero target value around which one maximizes the variance (Suykens et al., 2003). When choosing a good feature map, one expects the latent variables to be normally distributed around zero. This property resembles the added regularization term in the objective of the VAE (Kingma & Welling, 2014), which is expressed as the Kullback-Leibler divergence between the encoder's distribution and a unit Gaussian as a prior on the latent variables. **3)** Kernel PCA gives uncorrelated components in feature space. While it was already shown that PCA does not give a good disentangled representation for images (Eastwood & Williams, 2018; Higgins et al., 2017). Hence by designing a good kernel (through appropriate feature-maps) and doing kernel PCA, it is possible to get a disentangled representation for images as we show on the example in Figure 5. The uncorrelated components enhances the interpretation of the model.

## 4 The Gen-RKM Algorithm

Based on the previous analysis, we propose a novel algorithm, called the Gen-RKM algorithm, combining kernel learning and generative models. We show that this procedure is efficient to train and evaluate. It is also scalable to large datasets when using explicit feature maps. The training procedure simultaneously involves feature selection, common-subspace learning and pre-image map learning. This is achieved via an optimization procedure where one iteration involves an eigen-decomposition of the kernel matrix which is composed of the features from various views (see Eq. 3). The latent variables are given by the eigenvectors, which are then passed via a pre-image map to reconstruct the sample. Figure 1 shows a schematic representation of the algorithm when two data sources are available.

Thanks to training in $m$ mini-batches, this procedure is scalable to large datasets (sample size $N$) with training time scaling super-linearly with $T_m = c\frac{N^\gamma}{m^{\gamma-1}}$, instead of $T_k = cN^\gamma$, where $\gamma \approx 3$ for algorithms based on decomposition methods, with some proportionality constant $c$. The training time could be further reduced by computing the covariance matrix (size $(d_f + p_f) \times (d_f + p_f)$) instead

of a kernel matrix (size $\frac{N}{m} \times \frac{N}{m}$), when the sum of the dimensions of the feature-spaces is less than the samples in mini-batch i.e. $d_f + p_f \leq \frac{N}{m}$. While using neural networks as feature maps, $d_f$ and $p_f$ correspond to the number of neurons in the output layer, which are chosen as hyperparameters by the practitioner. Eigendecomposition of this smaller covariance matrix would yield $\boldsymbol{U}$ and $\boldsymbol{V}$ as eigenvectors (see Eq. 10 and Appendix A.2 for detailed derivation), where computing the $\boldsymbol{h}_i$ involves only matrix-multiplication which is readily parallelizable on modern GPUs:

$$
\begin{bmatrix} \frac{1}{\eta_1}\Phi_{\boldsymbol{x}}\Phi_{\boldsymbol{x}}^\top & \frac{1}{\eta_1}\Phi_{\boldsymbol{x}}\Phi_{\boldsymbol{y}}^\top \\ \frac{1}{\eta_2}\Phi_{\boldsymbol{y}}\Phi_{\boldsymbol{x}}^\top & \frac{1}{\eta_2}\Phi_{\boldsymbol{y}}\Phi_{\boldsymbol{y}}^\top \end{bmatrix} \begin{bmatrix} \boldsymbol{U} \\ \boldsymbol{V} \end{bmatrix} = \begin{bmatrix} \boldsymbol{U} \\ \boldsymbol{V} \end{bmatrix} \Lambda, \quad \begin{aligned} \Phi_{\boldsymbol{x}} &:= [\phi_1(\boldsymbol{x}_1), \dots, \phi_1(\boldsymbol{x}_N)], \\ \Phi_{\boldsymbol{y}} &:= [\phi_2(\boldsymbol{y}_1), \dots, \phi_2(\boldsymbol{y}_N)]. \end{aligned} \tag{10}
$$

---

**Algorithm 1** Gen-RKM

---

**Input:** $\{\boldsymbol{x}_i, \boldsymbol{y}_i\}_{i=1}^N$, $\eta_1$, $\eta_2$, feature map $\phi_j(\cdot)$ - explicit *or* implicit via kernels $k_j(\cdot, \cdot)$, for $j \in \{1, 2\}$
**Output:** Generated data $\boldsymbol{x}^\star$, $\boldsymbol{y}^\star$

1: **procedure** TRAIN
2:     **if** $\phi_j(\cdot)$ = Implicit **then**
3:         Hyperparameters: kernel specific
4:         Solve Eq. 3
5:         Select $s$ principal components
6:     **else if** $\phi_j(\cdot)$ = Explicit **then**
7:         **while** not converged **do**
8:             $\{\boldsymbol{x}, \boldsymbol{y}\} \leftarrow \{\text{Get mini-batch}\}$
9:             $\phi_1(\boldsymbol{x}) \leftarrow \boldsymbol{x}$; $\phi_2(\boldsymbol{y}) \leftarrow \boldsymbol{y}$
10:           do steps 4-5
11:           $\{\phi_1(\boldsymbol{x}), \phi_2(\boldsymbol{y})\} \leftarrow h$ (Eq. 6)
12:           $\{\boldsymbol{x}, \boldsymbol{y}\} \leftarrow \{\psi_1(\phi_1(\boldsymbol{x})), \psi_2(\phi_2(\boldsymbol{y}))\}$
13:           $\Delta\boldsymbol{\theta}_1 \propto -\nabla_{\boldsymbol{\theta}_1}\mathcal{J}_c$; $\Delta\boldsymbol{\theta}_2 \propto -\nabla_{\boldsymbol{\theta}_2}\mathcal{J}_c$
14:           $\Delta\boldsymbol{\zeta}_1 \propto -\nabla_{\boldsymbol{\zeta}_1}\mathcal{J}_c$; $\Delta\boldsymbol{\zeta}_2 \propto -\nabla_{\boldsymbol{\zeta}_2}\mathcal{J}_c$
15:         **end while**
16:     **end if**
17: **end procedure**

1: **procedure** GENERATION
2:     Select $\boldsymbol{h}^\star$
3:     **if** $\phi_j(\cdot)$ = Implicit **then**
4:         Hyperparameter: $n_r$
5:         Compute $\boldsymbol{k}_{\boldsymbol{x}^*}$, $\boldsymbol{k}_{\boldsymbol{y}^*}$ (Eq. 7)
6:         Get $\hat{\boldsymbol{x}}$, $\hat{\boldsymbol{y}}$ (Eq. 8)
7:     **else if** $\phi_j(\cdot)$ = Explicit **then**
8:         do steps 11-12
9:     **end if**
10: **end procedure**

---

## 5 EXPERIMENTS

To demonstrate the applicability of the proposed framework and algorithm, we trained the Gen-RKM model on a variety of datasets commonly used to evaluate generative models: MNIST (Le-Cun & Cortes, 2010), Fashion-MNIST (Xiao et al., 2017), CIFAR-10 (Krizhevsky, 2009), CelebA (Liu et al., 2015), Dsprites (Matthey et al., 2017) and Teapot (Eastwood & Williams, 2018). The experiments were performed using both the implicit feature map defined by a Gaussian kernel and parametric explicit feature maps defined by deep neural networks, either convolutional or fully connected. As explained in Section 2, in case of kernel methods, training only involves constructing the kernel matrix and solving the eigenvalue problem in Eq. 3. In our experiments, we fit a Gaussian mixture model (GMM) with $l$ components to the latent variables of the training set, and randomly sample a new point $\boldsymbol{h}^\star$ for generating views using a kernel smoother. In case of explicit feature maps, we define $\phi_{1_{\theta_1}}$ and $\psi_{1_{\zeta_1}}$ as convolution and transposed-convolution neural networks, respectively (Dumoulin & Visin, 2016); and $\phi_{2_{\theta_2}}$ and $\psi_{1_{\zeta_2}}$ as fully-connected networks. The particular architecture details are outlined in Table 3 in the Appendix. The training procedure in case of explicitly defined maps consists of minimizing $\mathcal{J}_c$ using the Adam optimizer (Kingma & Ba, 2014) to update the weights and biases. To speed-up learning, we subdivided the datasets into $m$ mini-batches, and within each iteration of the optimizer, Eq. 3 is solved to update the value of $\mathcal{H}$. Information on the datasets and hyperparameters used for the experiments is given in Table 4 in the Appendix.

**Generation:**

Qualitative examples: Figure 2 shows the generated images using a convolutional neural network and transposed-convolutional neural network as the feature map and pre-image map respectively. The first column in yellow-boxes shows the training samples and the second column on the right shows the reconstructed samples. The other images shown are generated by random sampling from

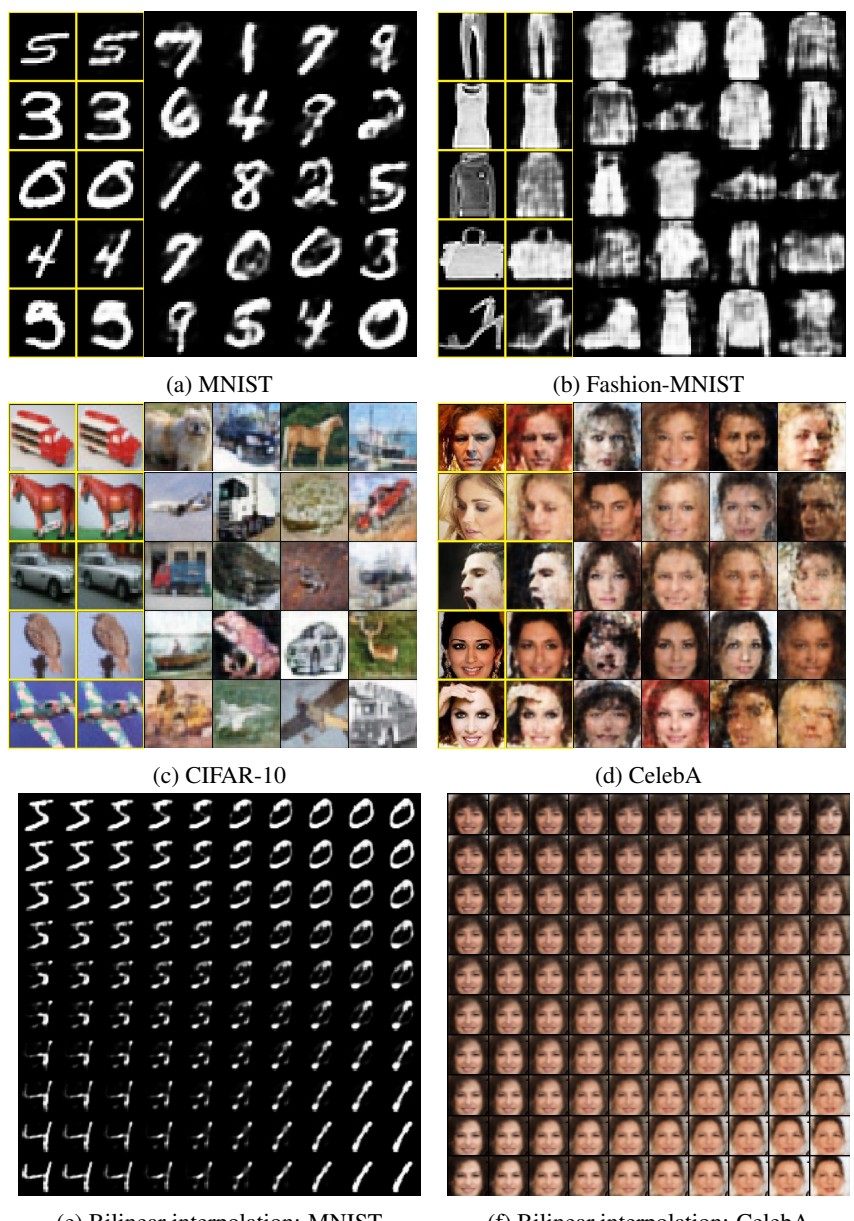

|     |     |
| --- | --- |
| (a) MNIST | (b) Fashion-MNIST |
| (c) CIFAR-10 | (d) CelebA |
| (e) Bilinear interpolation: MNIST | (f) Bilinear interpolation: CelebA |

Figure 2: Generated samples from the model using CNN as explicit feature map in the kernel function. In (a), (b), (c), (d) the yellow boxes in the first column show training examples and the adjacent boxes show the reconstructed samples. The other images (columns 3-6) are generated by random sampling from the fitted distribution over the learned latent variables. (e) and (f) shows the generated images through bilinear interpolations in the latent space.

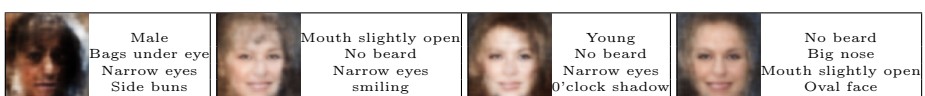

Figure 3: Multi-view generation on CelebA dataset showing images and attributes.

| 1 | 1 | 9 | 8 | 7 | 3 | 0 | 6 | 3 | 4 | 2 | 5 |

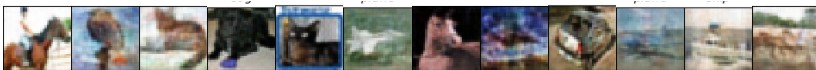

(a) MNIST: Implicit feature maps with Gaussian kernel are used during training. For generation, the pre-images are computed using the kernel-smoother method.

| 9 | 1 | 7 | 6 | 4 | 6 | 0 | 8 | 2 | 1 | 3 | 7 |

(b) MNIST: Explicit feature maps and the corresponding pre-image maps are defined by the Convolutional Neural Networks.

*horse bird horse dog cat plane horse bird car plane ship deer*

(c) CIFAR-10: Explicit feature maps as Convolutional Neural Networks. Pre-images are computed using Transposed CNNs.

Figure 4: Multi-view Generation (images and labels) on various datasets using implicit and explicit feature maps.

a GMM over the learned latent variables. Notice that the reconstructed samples are of better quality visually than the other images generated by random sampling. To elucidate that the model has not merely memorized the training examples, we show the generated images via bilinear-interpolations in the latent space in 2e and 2f.

Comparison: We compare the proposed model with the standard VAE (Kingma & Welling, 2014). For a fair comparison, the models have the same encoder/decoder architecture, optimization parameters and are trained until convergence, where the details are given in Table 3. We evaluate the performance qualitatively by comparing reconstruction and random sampling, the results are shown in Figure 8 in the Appendix. In order to quantitatively assess the quality of the randomly generated samples, we use the Fréchet Inception Distance (FID) introduced by Heusel et al. (2017). The results are reported in Table 1. Experiments were repeated for different latent-space dimensions ($h_{dim}$), and we observe empirically that FID scores are better for the Gen-RKM. This is confirmed by the qualitative evaluation in Table 8, where the VAE generates smoother images. An interesting trend could be noted that as the dimension of latent-space is increased, VAE gets better at generating images whereas the performance of Gen-RKM decreases slightly. This is attributed to the eigendecomposition of the kernel matrix whose eigenvalue spectrum decreases rapidly depicting that most information is captured in few principal components, while the rest is noise. The presence of noise hinders the convergence of the model. It is therefore important to select the number of latent variables proportionally to the size of the mini-batch and the corresponding spectrum of the kernel matrix (the diversity within a mini-batch affects the eigenvalue spectrum of the kernel matrix).

Table 1: FID Scores (Heusel et al., 2017) for randomly generated samples (smaller is better).

| Dataset | Algorithm | FID score | | |
|---|---|---|---|---|
| | | $h_{dim} = 10$ | $h_{dim} = 30$ | $h_{dim} = 50$ |
| MNIST | Gen-RKM | **89.825** | **130.497** | **131.696** |
| | VAE | 250 | 234.749 | 205.282 |
| CelebA | Gen-RKM | **103.299** | **84.403** | **85.121** |
| | VAE | 286.039 | 245.738 | 225.783 |

**Multi-view Generation:** Figures 3 & 4 demonstrate the multi-view generative capabilities of the model. In these datasets, labels or attributes are seen as another view of the image that provides extra information. One-hot encoding of the labels was used to train the model. Figure 4a shows the generated images and labels when feature maps are only implicitly known i.e. through a Gaussian kernel. Figures 4b, 4c shows the same when using fully-connected networks as parametric functions

to encode and decode labels. We can see that both the generated image and the generated label matches in most cases, albeit not all.

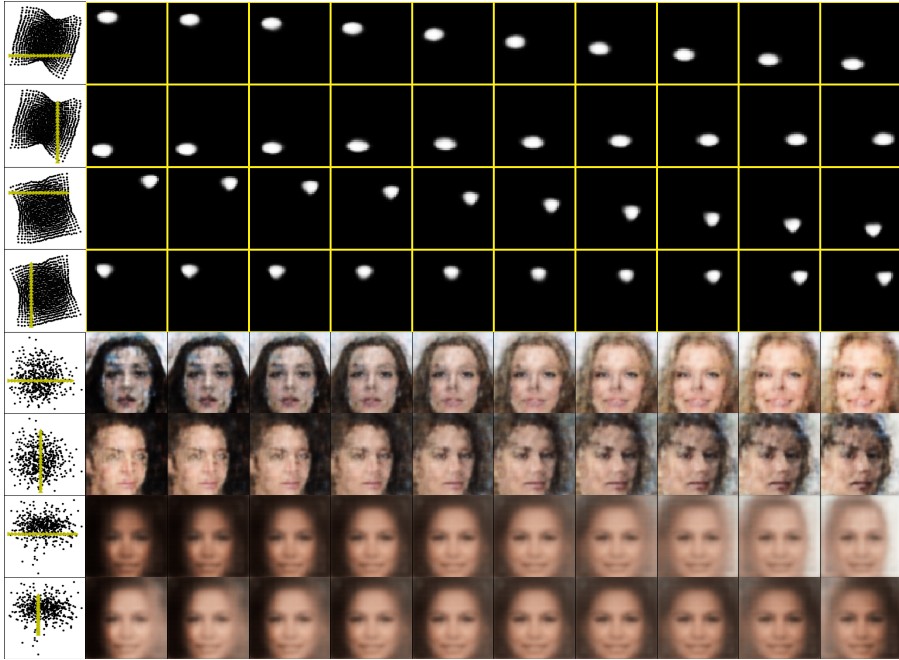

Figure 5: Exploring the learned uncorrelated-features by traversing along the eigenvectors. The first column shows the scatter plot of latent variables using the top two principal components. The green lines within, show the traversal in the latent space and the related rows show the corresponding reconstructed images.

Table 2: Disentanglement Metric on DSprites and Teapot dataset with Lasso and Random Forest regressor (Eastwood & Williams, 2018). For disentanglement and completeness higher score is better, for informativeness, lower is better.

| | $h_{dim}$ | Algorithm | Lasso | | | Random Forest | | |
|---|---|---|---|---|---|---|---|---|
| | | | Disent. | Comple. | Inform. | Disent. | Comple. | Inform. |
| DSprites | 10 | Gen-RKM | 0.30 | 0.10 | 0.87 | 0.12 | 0.10 | 0.28 |
| | | VAE | 0.11 | 0.09 | **0.17** | **0.73** | **0.54** | **0.06** |
| | | $\beta$-VAE $(\beta = 3)$ | **0.53** | **0.18** | 0.18 | 0.58 | 0.36 | **0.06** |
| | 2 | Gen-RKM | **0.72** | **0.71** | **0.64** | **0.05** | 0.19 | **0.03** |
| | | VAE | 0.04 | 0.01 | 0.87 | 0.01 | 0.13 | 0.11 |
| | | $\beta$-VAE $(\beta = 3)$ | 0.13 | 0.40 | 0.71 | 0.00 | **0.26** | 0.09 |
| Teapot | 10 | Gen-RKM | 0.28 | 0.23 | 0.39 | **0.48** | **0.39** | **0.19** |
| | | VAE | 0.28 | 0.21 | **0.36** | 0.30 | 0.27 | 0.21 |
| | | $\beta$-VAE $(\beta = 3)$ | **0.33** | **0.25** | **0.36** | 0.31 | 0.24 | 0.20 |
| | 5 | Gen-RKM | 0.22 | 0.23 | 0.74 | 0.08 | 0.09 | **0.27** |
| | | VAE | 0.16 | 0.14 | **0.66** | 0.11 | 0.14 | 0.28 |
| | | $\beta$-VAE $(\beta = 3)$ | **0.31** | **0.25** | 0.68 | **0.13** | **0.15** | 0.29 |

**Disentanglement:**

Qualitative examples: The latent variables are uncorrelated, which gives an indication that the model could resemble a disentangled representation. This is confirmed by the empirical evidence on Figure 5, where we explore the uncorrelated features learned by the models on the Dsprites and celebA dataset. In our experiments, the Dsprites training dataset comprised of $32 \times 32$ positions of oval and heart-shaped objects. The number of principal components chosen were 2 and the goal was to findout whether traversing along the eigenvectors, corresponds to traversing the generated im-

age in one particular direction while preserving the shape of the object. Rows 1 and 2 of Figure 5 show the reconstructed images of an oval while moving along first and second principal component respectively. Notice that the first and second components correspond to the $y$ and $x$ positions respectively. Rows 3 and 4 show the same for hearts. On the celebA dataset, we train the Gen-RKM with 15 components. Rows 5 and 6 shows the reconstructed images while traversing along the principal components. When moving along the first component from left-to-right, the hair-color of the women changes, while preserving the face structure. Whereas traversal along the second component, transforms a man to woman while preserving the orientation. When the number of principal components were 2 while training, the brightness and background light-source corresponds to the two largest variances in the dataset. Also notice that, the reconstructed images are more blurry due to the selection of less number of components to model $\mathcal{H}$.

Comparison: To quantitatively assess disentanglement performance, we compare Gen-RKM with VAE (Kingma & Welling, 2014) and beta-VAE (Higgins et al., 2017) on the Dsprites and Teapot datasets (Eastwood & Williams, 2018). The models have the same encoder/decoder architecture, optimization parameters and are trained until convergence, where the details are given in Table 3. The performance is measured using the proposed framework[3] of Eastwood & Williams (2018), which gives 3 measures: disentanglement, completeness and informativeness. The results are depicted in Table 2. Gen-RKM has good performance on the Dsprites dataset when the latent space dimension is equal to 2. This is expected as the number of disentangled generating factors in the dataset is also equal to 2, hence there are no noisy components in the kernel PCA hindering the convergence. The opposite happens in the case $h_{dim} = 10$, where noisy component are present. The above is confirmed by the Relative Importance Matrix on Figure 6 in the Appendix, where the 2 generating factors are well separated in the latent space of the Gen-RKM. For the Teapot dataset, Gen-RKM has good performance when $h_{dim} = 10$. More components are needed to capture all variations in the dataset, where the number of generating factors is now equal to 5. In the other cases, Gen-RKM has a performance comparable to the others.

## 6    CONCLUSION AND FUTURE WORK

The paper proposes a novel framework, called Gen-RKM, for generative models based on RKMs with extensions to multi-view generation and learning uncorrelated representations. This allows for a mechanism where the feature map can be implicitly defined using kernel functions or explicitly by (deep) neural network based methods. When using kernel functions, the training consists of only solving an eigenvalue problem. In the case of a (convolutional) neural network based explicit feature map, we used (transposed) networks as the pre-image functions. Consequently, a training procedure was proposed which involves joint feature-selection and subspace learning. Thanks to training in mini-batches and capability of working with covariance matrices, the training is scalable to large datasets. Experiments on benchmark datasets illustrate the merit of the proposed framework for generation quality as well as disentanglement. Extensions of this work consists of adapting the model to more advanced multi-view datatsets involving speech, images and texts; further analysis on other feature maps, pre-image methods, loss-functions and uncorrelated feature learning. Finally, this paper has demonstrated the applicability of the Gen-RKM framework, suggesting new research directions to be worth exploring.

---

[3]Code and dataset available at `https://github.com/cianeastwood/qedr`

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

## A APPENDIX

### A.1 DERIVATION OF GEN-RKM OBJECTIVE FUNCTION

Given $\mathcal{D} = \{\boldsymbol{x}_i, \boldsymbol{y}_i\}_{i=1}^N$, where $\boldsymbol{x}_i \in \mathbb{R}^d$, $\boldsymbol{y}_i \in \mathbb{R}^p$ and feature-map $\phi_1 : \mathbb{R}^d \mapsto \mathbb{R}^{d_f}$ and $\phi_2 : \mathbb{R}^p \mapsto \mathbb{R}^{p_f}$, the Least-Squares Support Vector Machine (LS-SVM) formulation of Kernel PCA (Suykens et al., 2002) for the two data sources can be written as:

$$\min_{\boldsymbol{U},\boldsymbol{V},\boldsymbol{e}_i} \frac{\eta_1}{2} \operatorname{Tr}(\boldsymbol{U}^\top \boldsymbol{U}) + \frac{\eta_2}{2} \operatorname{Tr}(\boldsymbol{V}^\top \boldsymbol{V}) - \frac{1}{2\lambda} \sum_{i=1}^N \boldsymbol{e}_i^\top \boldsymbol{e}_i \tag{11}$$

$$\text{s.t. } \boldsymbol{e}_i = \boldsymbol{U}^\top \phi_1(\boldsymbol{x}_i) + \boldsymbol{V}^\top \phi_2(\boldsymbol{y}_i) \quad \forall i = 1, \dots, N,$$

where $\boldsymbol{U} \in \mathbb{R}^{d \times s}$ and $\boldsymbol{V} \in \mathbb{R}^{p \times s}$ are the interconnection matrices.

Using the notion of *conjugate feature duality* introduced in Suykens (2017), the error variables $\boldsymbol{e}_i$ are conjugated to latent variables $\boldsymbol{h}_i$ using:

$$\frac{1}{2\lambda} \boldsymbol{e}^\top \boldsymbol{e} + \frac{\lambda}{2} \boldsymbol{h}^\top \boldsymbol{h} \geq \boldsymbol{e}^\top \boldsymbol{h}, \qquad \forall \boldsymbol{e}, \boldsymbol{h} \in \mathbb{R}^s \tag{12}$$

which is also known as the Fenchel-Young inequality for the case of quadratic functions (Rockafellar, 1974). By eliminating the variables $\boldsymbol{e}_i$ from Eq. 11 and using Eq. 12, we obtain the Gen-RKM training objective function:

$$\mathcal{J}_t = \sum_{i=1}^N \left( -\phi_1(\boldsymbol{x}_i)^\top \boldsymbol{U} \boldsymbol{h}_i - \phi_2(\boldsymbol{y}_i)^\top \boldsymbol{V} \boldsymbol{h}_i + \frac{\lambda}{2} \boldsymbol{h}_i^\top \boldsymbol{h}_i \right) + \frac{\eta_1}{2} \operatorname{Tr}(\boldsymbol{U}^\top \boldsymbol{U}) + \frac{\eta_2}{2} \operatorname{Tr}(\boldsymbol{V}^\top \boldsymbol{V}). \tag{13}$$

### A.2 KERNEL PCA IN THE PRIMAL

From Eq. 2, eliminating the variables $\boldsymbol{h}_i$ yields the following:

$$\frac{1}{\eta_1} \left[ \sum_{i=1}^N \phi_1(\boldsymbol{x}_i)\phi_1(\boldsymbol{x}_i)^\top \boldsymbol{U} + \sum_{i=1}^N \phi_1(\boldsymbol{x}_i)\phi_2(\boldsymbol{y}_i)^\top \boldsymbol{V} \right] = \lambda \boldsymbol{U},$$

$$\frac{1}{\eta_2} \left[ \sum_{i=1}^N \phi_2(\boldsymbol{y}_i)\phi_1(\boldsymbol{x}_i)^\top \boldsymbol{U} + \sum_{i=1}^N \phi_2(\boldsymbol{y}_i)\phi_2(\boldsymbol{y}_i)^\top \boldsymbol{V} \right] = \lambda \boldsymbol{V}. \tag{14}$$

Denote $\Phi_{\boldsymbol{x}} := [\phi_1(\boldsymbol{x}_1), \dots, \phi_1(\boldsymbol{x}_N)]$, $\Phi_{\boldsymbol{y}} := [\phi_2(\boldsymbol{y}_1), \dots, \phi_2(\boldsymbol{y}_N)]$ and $\boldsymbol{\Lambda} = \operatorname{diag}\{\lambda_1, \dots, \lambda_s\} \in \mathbb{R}^{s \times s}$ with $s \leq N$. Now, composing the above equations in matrix form, we get the following eigen-decomposition problem:

$$\begin{bmatrix} \frac{1}{\eta_1} \Phi_{\boldsymbol{x}} \Phi_{\boldsymbol{x}}^\top & \frac{1}{\eta_1} \Phi_{\boldsymbol{x}} \Phi_{\boldsymbol{y}}^\top \\ \frac{1}{\eta_2} \Phi_{\boldsymbol{y}} \Phi_{\boldsymbol{x}}^\top & \frac{1}{\eta_2} \Phi_{\boldsymbol{y}} \Phi_{\boldsymbol{y}}^\top \end{bmatrix} \begin{bmatrix} \boldsymbol{U} \\ \boldsymbol{V} \end{bmatrix} = \begin{bmatrix} \boldsymbol{U} \\ \boldsymbol{V} \end{bmatrix} \boldsymbol{\Lambda}. \tag{15}$$

Here the size of the covariance matrix is $(d_f + p_f) \times (d_f + p_f)$. The latent variables $\boldsymbol{h}_i$ can be computed using Eq. 2, which simply involves matrix multiplications.

### A.3 STABILIZING THE OBJECTIVE FUNCTION

**Proposition 1.** *All stationary solutions for $\boldsymbol{H}, \boldsymbol{\Lambda}$ in Eq. 3 of $\mathcal{J}_t$ lead to $\mathcal{J}_t = 0$.*

*Proof.* Let $\lambda_i, \boldsymbol{h}_i$ are given by Eq. 3. Using Eq. 2 to substitute $\boldsymbol{V}$ and $\boldsymbol{U}$ in Eq. 1 yields:

$$\mathcal{J}_t(\boldsymbol{V}, \boldsymbol{U}, \boldsymbol{\Lambda}, \boldsymbol{H}) = \sum_{i=1}^N -\frac{\lambda}{2} \boldsymbol{h}_i^\top \boldsymbol{h}_i + \frac{\eta_1}{2} \operatorname{Tr} \left( \frac{1}{\eta_1^2} \sum_{i=1}^N \boldsymbol{h}_i \phi_1(\boldsymbol{x}_i)^\top \sum_{j=1}^N \phi_1(\boldsymbol{x}_j) \boldsymbol{h}_j^\top \right)$$

$$+ \frac{\eta_2}{2} \operatorname{Tr} \left( \frac{1}{\eta_2^2} \sum_{i=1}^N \boldsymbol{h}_i \phi_2(\boldsymbol{y}_i)^\top \sum_{j=1}^N \phi_2(\boldsymbol{y}_j) \boldsymbol{h}_j^\top \right)$$

$$= \sum_{i=1}^{N} -\frac{\lambda}{2} \boldsymbol{h}_i^\top \boldsymbol{h}_i + \frac{\eta_1}{2} \operatorname{Tr}\left(\frac{1}{\eta_1^2} \boldsymbol{H} \boldsymbol{K}_1 \boldsymbol{H}^\top\right) + \frac{\eta_2}{2} \operatorname{Tr}\left(\frac{1}{\eta_2^2} \boldsymbol{H} \boldsymbol{K}_2 \boldsymbol{H}^\top\right)$$

$$= \sum_{i=1}^{N} -\frac{\lambda}{2} \boldsymbol{h}_i^\top \boldsymbol{h}_i + \frac{1}{2} \operatorname{Tr}\left(\boldsymbol{H} \left[\frac{1}{\eta_1} \boldsymbol{K}_1 + \frac{1}{\eta_2} \boldsymbol{K}_2\right] \boldsymbol{H}^\top\right).$$

From Eq. 3, we get:

$$\mathcal{J}_t(\boldsymbol{V}, \boldsymbol{U}, \boldsymbol{\Lambda}, \boldsymbol{H}) = \sum_{i=1}^{N} -\frac{\lambda}{2} \boldsymbol{h}_i^\top \boldsymbol{h}_i + \frac{1}{2} \operatorname{Tr}\left(\boldsymbol{H} \boldsymbol{H}^\top \lambda\right) = \sum_{i=1}^{N} -\frac{\lambda}{2} \boldsymbol{h}_i^\top \boldsymbol{h}_i + \frac{\lambda}{2} \sum_{i=1}^{N} \boldsymbol{h}_i^\top \boldsymbol{h}_i = 0.$$

$\square$

**Proposition 2.** *Let $J(\boldsymbol{x}) : \mathbb{R}^N \to \mathbb{R}$ be a smooth function, for all $\boldsymbol{x} \in \mathbb{R}^N$ and for $c \in \mathbb{R}_{>0}$, define $\bar{J}(\boldsymbol{x}) := J(\boldsymbol{x}) + \frac{c}{2} J(\boldsymbol{x})^2$. Assuming $(1 + cJ(\boldsymbol{x})) \neq 0$, then $\boldsymbol{x}^\star$ is the stationary points of $\bar{J}(\boldsymbol{x})$ iff $\boldsymbol{x}^\star$ is the stationary point for $J(\boldsymbol{x})$.*

*Proof.* Let $\boldsymbol{x}^\star$ be a stationary point of $J(\boldsymbol{x})$, meaning that $\nabla J(\boldsymbol{x}^\star) = 0$. The stationary points for $\bar{J}(\boldsymbol{x})$ can be obtained from:

$$\frac{d\bar{J}}{d\boldsymbol{x}} = (\nabla J(\boldsymbol{x}) + cJ(\boldsymbol{x})\nabla J(\boldsymbol{x})) = (1 + cJ(\boldsymbol{x})) \nabla J(\boldsymbol{x}). \tag{16}$$

It is easy to see from Eq. 2 that if $\boldsymbol{x} = \boldsymbol{x}^*$, $\nabla J(\boldsymbol{x}^*) = 0$, we have that $\frac{d\bar{J}}{d\boldsymbol{x}}\Big|_{\boldsymbol{x}^*} = 0$, meaning that all the stationary points of $J(\boldsymbol{x})$ are stationary points of $\bar{J}(\boldsymbol{x})$.

To show the other way, let $\boldsymbol{x}^\star$ be stationary point of $\bar{J}(\boldsymbol{x})$ i.e. $\nabla \bar{J}(\boldsymbol{x}^\star) = 0$. Assuming $(1 + cJ(\boldsymbol{x}^\star)) \neq 0$, then from Eq. 16 for all $c \in \mathbb{R}_{>0}$, we have

$$(1 + cJ(\boldsymbol{x}^\star)) \nabla J(\boldsymbol{x}^\star) = 0,$$

implying that $\nabla J(\boldsymbol{x}^\star) = 0$.

$\square$

Based on the above propositions, we stabilize our original objective function Eq. 1 to keep it bounded and hence is suitable for minimization with Gradient-descent methods. Without the reconstruction errors, the stabilized objective function is

$$\min_{\boldsymbol{U}, \boldsymbol{V}, \boldsymbol{h}_i} \mathcal{J}_t + \frac{c}{2} \mathcal{J}_t^2.$$

Denoting $\bar{J} = \mathcal{J}_t + \frac{c_{stab}}{2} \mathcal{J}_t^2$. Since the derivatives of $\mathcal{J}_t$ are given by Eq. 2, the stationary points of $\bar{J}$ are:

$$\begin{cases} \frac{\partial \bar{J}}{\partial \boldsymbol{V}} = (1 + c_{stab} \mathcal{J}_t) \left(-\sum_{i=1}^{N} \phi_1(\boldsymbol{x}_i) \boldsymbol{h}_i^\top + \eta_1 \boldsymbol{V}\right) = 0 & \implies \boldsymbol{V} = \frac{1}{\eta_1} \sum_{i=1}^{N} \phi_1(\boldsymbol{x}_i) \boldsymbol{h}_i^\top, \\ \frac{\partial \bar{J}}{\partial \boldsymbol{U}} = (1 + c_{stab} \mathcal{J}_t) \left(-\sum_{i=1}^{N} \phi_2(\boldsymbol{y}_i) \boldsymbol{h}_i^\top + \eta_2 \boldsymbol{U}\right) = 0 & \implies \boldsymbol{U} = \frac{1}{\eta_2} \sum_{i=1}^{N} \phi_2(\boldsymbol{y}_i) \boldsymbol{h}_i^\top, \\ \frac{\partial \bar{J}}{\partial \boldsymbol{h}_i} = (1 + c_{stab} \mathcal{J}_t) \left(-\boldsymbol{V}^\top \phi_1(\boldsymbol{x}_i) - \boldsymbol{U}^\top \phi_2(\boldsymbol{y}_i) + \lambda \boldsymbol{h}_i\right) = 0 & \implies \begin{aligned} \lambda \boldsymbol{h}_i &= \boldsymbol{V}^\top \phi_1(\boldsymbol{x}_i) \\ &+ \boldsymbol{U}^\top \phi_2(\boldsymbol{y}_i), \end{aligned} \end{cases}$$

assuming $1 + c_{stab} \mathcal{J}_t \neq 0$. Elimination of $\boldsymbol{V}$ and $\boldsymbol{U}$ yields $\left[\frac{1}{\eta_1} \boldsymbol{K}_1 + \frac{1}{\eta_2} \boldsymbol{K}_2\right] \boldsymbol{H}^\top = \boldsymbol{H}^\top \boldsymbol{\Lambda}$, which is indeed the same solution for $c_{stab} = 0$ in Eq. 1 and Eq. 3.

### A.4 CENTERING OF KERNEL MATRIX

Centering of the kernel matrix is done by the following equation:

$$\boldsymbol{K}_c = \boldsymbol{K} - N^{-1} \boldsymbol{1} \boldsymbol{1}^\top \boldsymbol{K} - N^{-1} \boldsymbol{K} \boldsymbol{1} \boldsymbol{1}^\top + N^{-2} \boldsymbol{1} \boldsymbol{1}^\top \boldsymbol{K} \boldsymbol{1} \boldsymbol{1}^\top, \tag{17}$$

where $\boldsymbol{1}$ denotes an $N$-dimensional vector of ones and $\boldsymbol{K}$ is either $\boldsymbol{K}_1$ or $\boldsymbol{K}_2$.

## A.5 ARCHITECTURE DETAILS

See Table 3 and 4 for details on model architectures, datasets and hyperparameters used in this paper. The PyTorch library in Python was used as the programming language with a 8GB NVIDIA QUADRO P4000 GPU.

Table 3: Details of model architectures used in the paper. All convolutions and transposed-convolutions are with stride 2 and padding 1. Unless stated otherwise, the layers have Parametric-RELU ($\alpha = 0.2$) activation function, except the output layers of the pre-image maps which has sigmoid activation function.

| Dataset | Optimizer (Adam) | | Architecture $\mathcal{X}$ | $\mathcal{Y}$ |
|---|---|---|---|---|
| MNIST | 1e-3 | Input | 28x28x1 | 10 (One-hot encoding) |
| | | Feature-map (fm) | Conv 32x4x4; Conv 64x4x4; FC 128 (Linear) | FC 15, 20 (Linear) |
| | | Pre-image map | reverse of fm | reverse of fm |
| | | Latent space dim. | | 500 |
| Fashion -MNIST | 1e-3 | Input | 28x28x1 | 10 (One-hot encoding) |
| | | Feature-map | Conv 32x4x4; 64x4x4; FC 128 (Linear) | FC 15, 20 |
| | | Pre-image map (fm) | reverse of fm | reverse of fm |
| | | Latent space dim. | | 100 |
| CIFAR-10 | 1e-3 | Input | 32x32x3 | 10 (One-hot encoding) |
| | | Feature-map (fm) | Conv 64x4x4; Conv 128x4x4; FC 128 (Linear) | FC 15, 20 |
| | | Pre-image map | reverse of fm | reverse of fm |
| | | Latent space dim. | | 500 |
| CelebA | 1e-4 | Input | 64x64x3 | - |
| | | Feature-map (fm) | Conv 32x4x4; Conv 64x4x4; Conv 128x4x4; Conv 256x4x4 ; FC 128 (Linear) | - |
| | | Pre-image map | reverse of fm | - |
| | | Latent space dim. | | 15 |
| Dsprites | 1e-4 | Input | 64x64x1 | - |
| | | Feature-map (fm) | Conv 20x4x4; Conv 40x4x4; Conv 80x4x4; FC 128 (Linear) | - |
| | | Pre-image map | reverse of fm | - |
| | | Latent space dim. | | 2/10 |
| Teapot | 1e-4 | Input | 64x64x3 | - |
| | | Feature-map (fm) | Conv 30x4x4; Conv 60x4x4; Conv 90x4x4; FC 128 (Linear) | - |
| | | Pre-image map | reverse of fm | - |
| | | Latent space dim. | | 5/10 |

Table 4: Datasets and hyperparameters used for the experiments. The bandwidth of the Gaussian kernel for generation corresponds to the bandwidth that gave the best performance determined by cross-validation on the MNIST classification problem.

| Dataset | $N$ | $d$ | $N_{\text{subset}}$ | $s$ | $m$ | $\sigma$ | $n_r$ | $l$ |
|---|---|---|---|---|---|---|---|---|
| MNIST | 60000 | $28 \times 28$ | 5000 | 500 | 50 | 1.3 | 4 | 10 |
| Fashion-MNIST | 60000 | $28 \times 28$ | 500 | 100 | 5 | / | / | 10 |
| CIFAR-10 | 60000 | $32 \times 32 \times 3$ | 500 | 500 | 5 | / | / | 10 |
| CelebA | 202599 | $128 \times 128 \times 3$ | 500 | 15 | 5 | / | / | 20 |
| Dsprites | 737280 | $64 \times 64$ | 1024 | 2/10 | 5 | / | / | / |
| Teapot | 200000 | $64 \times 64 \times 3$ | 1000 | 5/10 | 100 | / | / | / |

## A.6 BILINEAR INTERPOLATION

Given four vectors $\boldsymbol{h}_1, \boldsymbol{h}_2, \boldsymbol{h}_3$ and $\boldsymbol{h}_4$ (reconstructed images from these vectors are shown at the edges of Figs. 2e, 2f), the interpolated vector $\boldsymbol{h}^\star$ is given by:

$$\boldsymbol{h}^\star = (1-\alpha)(1-\gamma)\boldsymbol{h}_1 + \alpha(1-\gamma)\boldsymbol{h}_2 + \gamma(1-\alpha)\boldsymbol{h}_3 + \gamma\alpha\boldsymbol{h}_4, \quad 0 \le \alpha, \gamma \le 1.$$

This $\boldsymbol{h}^\star$ is then used in step 8 of the generation procedure of Gen-RKM algorithm (see Algorithm 1) to compute $\boldsymbol{x}^\star$.

## A.7 VISUALIZING THE DISENTANGLEMENT METRIC

In this section we show the Hinton plots to visualize the disentaglement scores as shown in Table 2. Following the conventions of Eastwood & Williams (2018), $\boldsymbol{z}$ represents the ground-truth data generating factors. Figs. 6 & 7 shows the Hinton plots on DSprites and Teapot datasets using Lasso and Random Forest regressors for various algorithms. Here the square size indicates the magnitude of the *relative importance* of the latent code $h_i$ in predicting $z_i$.

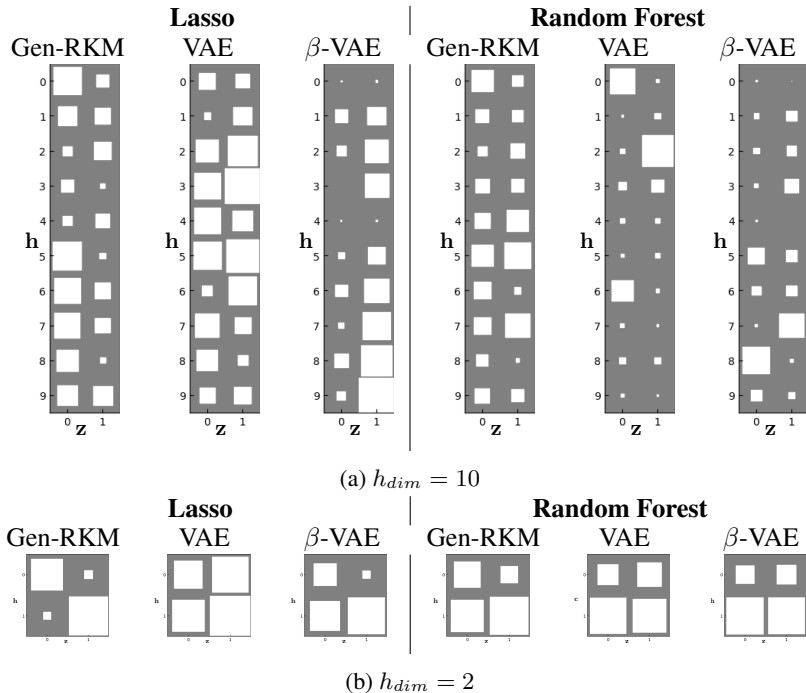

(a) $h_{dim} = 10$

(b) $h_{dim} = 2$

Figure 6: Relative importance matrix as computed by Lasso and Random Forest regaressors on DSprites dataset for $h_{dim} = \{10, 2\}$ against the underlying data generating factors $z_{dim} = \{2\}$ corresponding to $x, y$ positions of object.

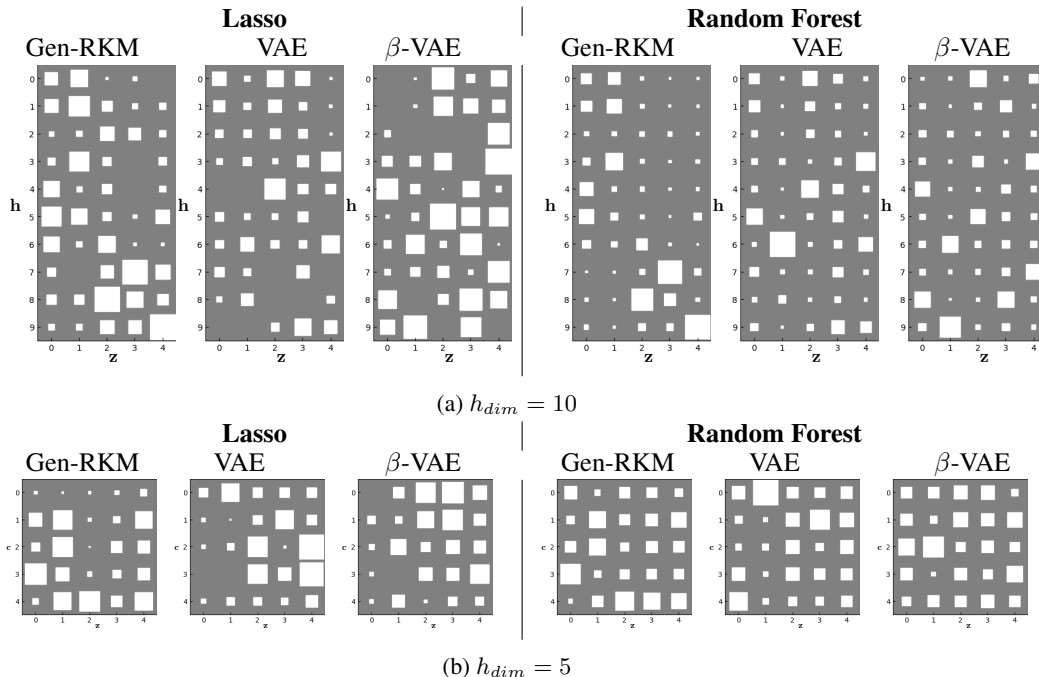

(a) $h_{dim} = 10$

(b) $h_{dim} = 5$

Figure 7: Relative importance matrix as computed by Lasso and Random Forest regaressors on Teapot dataset for $h_{dim} = \{10, 5\}$ against the underlying data generating factors $z_{dim} = \{5\}$ corresponding to azimuth, elevation and colors red, green and blue of the teapot object.

## A.8 FURTHER EMPIRICAL RESULTS

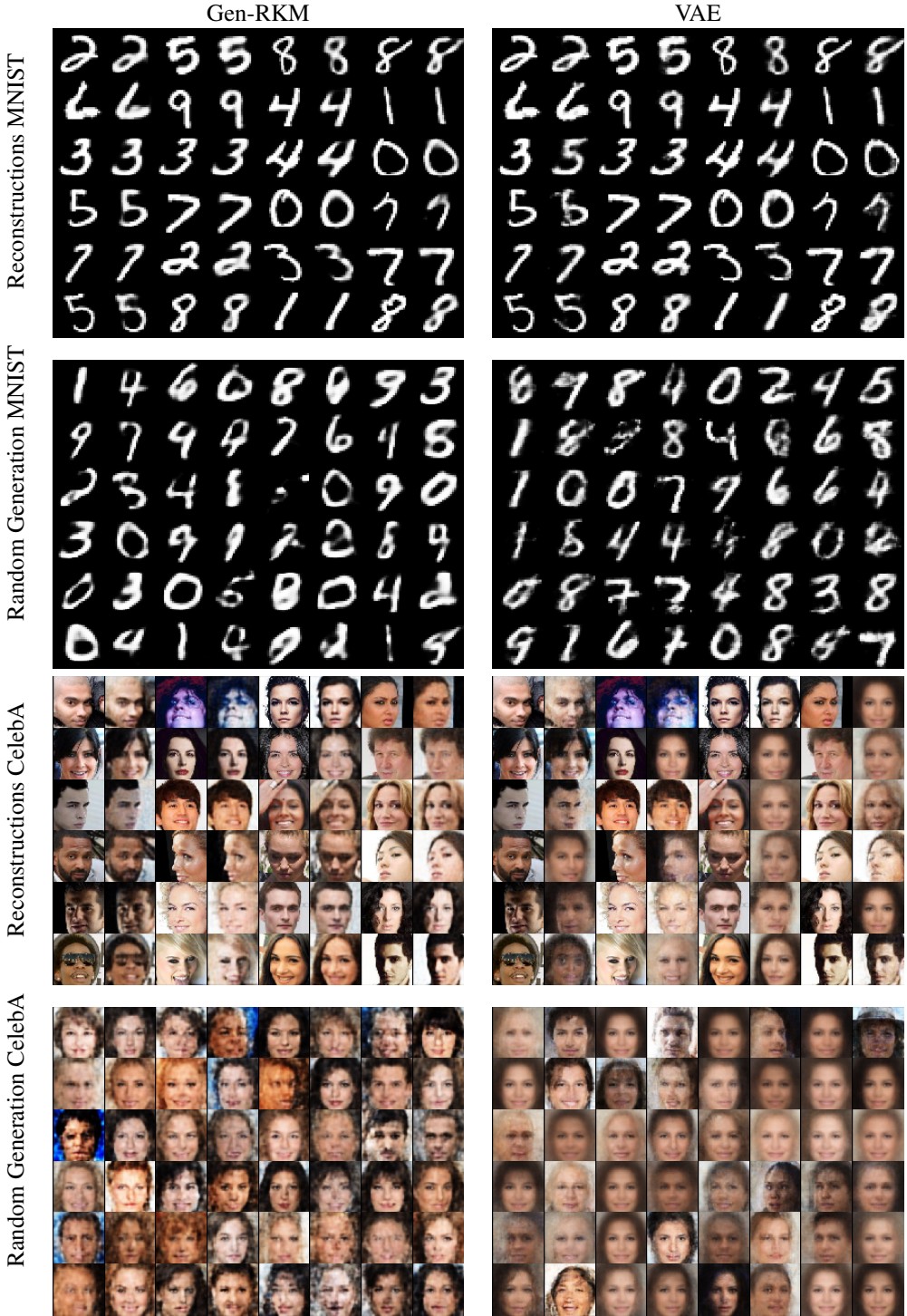

Figure 8: Comparing Gen-RKM and standard VAE for reconstruction and generation quality. In reconstruction MNIST and reconstruction CelebA, uneven columns correspond to the original image, even columns to the reconstructed image.

