# OpenReview forum: "Generative Restricted Kernel Machines"
_ICLR.cc/2020/Conference — Reject_

### Official Review · AnonReviewer1 · 2019-10-23
**Official Blind Review #1**

**Rating:** 3

**Review:**

This paper presents a model and training framework for generating samples based on restricted kernel machines. It is extended to multi-view generation and uncorrelated feature representation learning.

- The paper is well-written and well-organized. Notations and claims are clear.

- The idea of a multi-view generation model based on restricted kernel machines is interesting. However, the paper seems to be limited to model definition and algorithm overview without a performance evaluating analysis.


- The experimental evaluations are not satisfactory. Although it is claimed in the paper that the model is able to generate high quality images, it is very hard to be confirmed with these experiments. There is no concrete attempt at comparing the performance of the model to the other used methodologies. Generating high quality images with multiple views is an interesting problem, and there are good works in the field addressing the issues. To name a few:
Zhu, Z., Luo, P., Wang, X., Tang, X.: Multi-view perceptron: a deep model for learning face identity and view representations. In: Advances in Neural Information Processing Systems (NIPS). pp. 217–225, (2014)
Kan, M., Shan, S., Chen, X.: Multi-view deep network for cross-view classification. In: Proceedings of the IEEE Conference on Computer Vision and Pattern Recognition (CVPR). pp. 4847–4855, (2016)
Yin, X., Liu, X.: Multi-task convolutional neural network for pose-invariant face recognition. IEEE Transactions on Image Processing (2017)
Yim, J., Jung, H., Yoo, B., Choi, C., Park, D., Kim, J.: Rotating your face using multitask deep neural network. In: Proceedings of the IEEE Conference on Computer Vision and Pattern Recognition (CVPR). pp. 676–684, (2015)
Yu Tian, Xi Peng , Long Zhao, Shaoting Zhang , ,Dimitris N. Metaxas , CR-GAN: Learning Complete Representations for Multi-view Generation, arXiv: 1806.11191, 2018.

There might be differences between these works and the paper, it is common to evaluate the quality of the generation to other models in terms of accuracy or in the classification tasks. Unfortunately, there is no such quantitative analysis in the paper. So the advantages of the proposed model is not very clear since there is not enough quantitative performance analysis. It would be interesting to see complexity analysis to evaluate the computational costs.

Overall, I do not recommend this paper for publication. The experimental results are not satisfactory, and the paper needs improvements in that regard.

** update:
I would like to thank the authors for their comments. However, I still see major issues in the paper unresolved and my review remains the same.

**Experience Assessment:**

I do not know much about this area.

**Review Assessment: Checking Correctness Of Derivations And Theory:**

I assessed the sensibility of the derivations and theory.

**Review Assessment: Checking Correctness Of Experiments:**

I assessed the sensibility of the experiments.

**Review Assessment: Thoroughness In Paper Reading:**

I read the paper at least twice and used my best judgement in assessing the paper.

---

> ### Author Response · Authors · 2019-11-15
> **Response reviewer 1**
>
> We’d like to thank the reviewer for their review and helpful suggestions.
>
> 1) “The experimental evaluations are not satisfactory. Although it is claimed in the paper that the model is able to generate high quality images, it is very hard to be confirmed with these experiments. There is no concrete attempt at comparing the performance of the model to the other used methodologies.”
>
> We thank the reviewer for the feedback and agree with the statement . We extended the experimental section with additional experiments. The details are discussed in point 5 in the response to Reviewer 4.
>
>
> 2) “ There might be differences between these works and the paper, it is common to evaluate the quality of the generation to other models in terms of accuracy or in the classification tasks. Unfortunately, there is no such quantitative analysis in the paper.”
>
> Using the classification accuracy to assess the performance is an interesting approach. This however requires delicate fine-tuning of the parameters of the classifier and generative models. We therefore feel that other metrics are more suitable (see above comment) and leave the evaluation of the classification performance for future work as it requires a seperate in-depth study.
>
> 3) “It would be interesting to see complexity analysis to evaluate the computational costs.”
>
> We discuss the computational costs of the algorithm in Section 4. Scalability is known to be an issue for kernel PCA, where the SVD has a complexity O(n^3) with n the size of the dataset. This was hedged using mini-batches, which results in a complexity O(m^3) with m the size of the mini-batch. If the size of the mini-batch is still too large, we propose to use the covariance matrix instead of the kernel matrix, the complexity is now O(d^3), with d the size of the final layer of the feature map. If both the mini-batch size and the final layer are large, the method is rather slow (still cubic complexity).
>
> We hope this addresses the reviewer’s concerns.

---

### Official Review · AnonReviewer2 · 2019-10-24
**Official Blind Review #2**

**Rating:** 3

**Review:**

This is a very good paper, building on the idea of Restricted Kernel Machines (drawing a nice parallel between Restricted Bolzman Machines and tools available in the Kernel modelling literature). In this manuscript, the author(s) extend the work to a generative model setting to achieve multi-view generation -- a generative model that can explain correlated variables from a common subspace. The manuscript is well-written and easy to follow and the algorithmic details are clear. Image generation is illustrated on standard datasets (MNIST / CIFAR / CelebA).
While the framework and learning algorithm are good, and novel extensions to what appears to be previous work of the authors, I am less persuaded by the empirical work. Latent variable-based generative modes such as this (and this is motivated in the introduction to the paper) should be judged on if they can extract anything useful about the problem domain in the latent representations that we can interpret. This is not the case here --  the results presented are examples of images that the models can generate. No critical appraisal is given about when the models might fail or when one ought to resort to this approach and not a sample from the plethora of variants of VAE we read about. What have we learnt about images / hand-written characters / faces of popular people from a study like this?
From the above empirical results point of view, I do not think this manuscript is ready for publication, despite what I see as the elegance of the framework.

**Experience Assessment:**

I have read many papers in this area.

**Review Assessment: Checking Correctness Of Derivations And Theory:**

I assessed the sensibility of the derivations and theory.

**Review Assessment: Checking Correctness Of Experiments:**

I assessed the sensibility of the experiments.

**Review Assessment: Thoroughness In Paper Reading:**

I read the paper at least twice and used my best judgement in assessing the paper.

---

> ### Author Response · Authors · 2019-11-15
> **Response reviewer 2**
>
> We’d like to thank the reviewer for their review and helpful suggestions.
>
> 1) “This is a very good paper, building on the idea of Restricted Kernel Machines (drawing a nice parallel between Restricted Bolzman Machines and tools available in the Kernel modelling literature). In this manuscript, the author(s) extend the work to a generative model setting to achieve multi-view generation -- a generative model that can explain correlated variables from a common subspace. The manuscript is well-written and easy to follow and the algorithmic details are clear. Image generation is illustrated on standard datasets (MNIST / CIFAR / CelebA).”
>
> We thank you for the appreciation.
>
> 2) “Latent variable-based generative modes such as this (and this is motivated in the introduction to the paper) should be judged on if they can extract anything useful about the problem domain in the latent representations that we can interpret.”
>
> The interpretation of the latent space was addressed in point 3 in the response to Reviewer 4.
>
> 3) “When one ought to resort to this approach and not a sample from the plethora of variants of VAE we read about.”
>
> The comparison between Gen-RKM and the different VAE variants was done in point 4 in the response to Reviewer 4.
>
> 4) “No critical appraisal is given about when the models might fail”
>
> 4.1) Scalability towards large datasets could be an issue as the computational complexity of the SVD is O(n^3) with n the size of the dataset. This was hedged using mini-batches, which results in a complexity O(m^3) with m the size of the mini-batch. If the size of the mini-batch is still too large, we propose to use the covariance matrix instead of the kernel matrix, the complexity is now O(d^3), with d the size of the final layer of the feature map. If both the mini-batch size and the final layer are large, the method is rather slow (still cubic complexity).
> 4.2) The model has difficulties converging when the eigenvalue spectrum of the kernel matrix decreases rapidly, which means that most information is captured in a few principal components, while the rest of the components are noise. The presence of this noise hinders the convergence of the method as these eigenvectors can change drastically without affecting the reconstruction loss (small eigenvalue). It is therefore important to select the number of latent variables in proportion with the size of the mini-batch and the corresponding spectrum of the kernel matrix (the diversity within a mini-batch affects the eigenvalue spectrum of the kernel matrix).
>
> 5) “From the above empirical results point of view, I do not think this manuscript is ready for publication, despite what I see as the elegance of the framework.”
>
> We agree with the comment and extended the experimental section with multiple comparisons. The details are discussed in point 5 in the response to Reviewer 4.
>
> We hope this addresses the reviewer’s concerns.

---

### Official Review · AnonReviewer4 · 2019-10-28
**Official Blind Review #4**

**Rating:** 6

**Review:**

There exists two papers:
[1] Multimodal Learning with Deep Boltzmann Machines, http://jmlr.org/papers/volume15/srivastava14b/srivastava14b.pdf
[2] Deep Restricted Kernel Machines Using Conjugate Feature Duality, ftp://ftp.esat.kuleuven.ac.be/stadius/suykens/reports/deepRKM1.pdf

In particular [2] considers a model, which is similar to a Boltzmann machine, but at the same time it is based on kernel features, and uses structure of the corresponding optimization problem to obtain a solution in a semi-explicit way.

The authors of the considered paper
1) generalise a multimodal variant of the Boltzmann machine from [1] (which uses a special cross-product term to take into account dependency between modalities) to the case of kernel machines,
2) demonstrate on several typical datasets that using explicit deep network features it is possible to model images and data with two modalities (faces/textual description of faces).

Comments:
- From the description of the functionals L1 and L2 (bottom of page 4 and top of the page 5) the reader can think that the authors tune parameters (zeta_1,theta_1) and (zeta_2,theta_2) for each sample point separately
- The authors claimed that the experiments were done both for kernel features and for explicit features based on neural networks. However, in the experimental section there are no results obtained when using implicit kernels. Nothings is told on how to select kernel parameters
- The authors claimed that thanks to PCA-like definition of latent vectors they are orthogonal which is similar to disentangle representations. However, there are no any empirical evidences whether it is possible to benefit somehow from that orthogonal property, as well as there is no comparison with approaches to construct disentangleв latent representation for other types of generative models.

Conclusions:
- In general the text is accurately written, the work is well organised.
- Still I was not able to understand the main idea of the paper.
a) if the main idea of the paper that the authors propose some new method for generative modeling of multi-modal data, then the authors should make significantly more diverse experiments and ablation studies. Actually, this is not the case of the current work. The authors did not provide any quantitate measure and comparison with existing approaches;
b) if the main idea is to present a new approach, then still I would not call the approach completely new, as it is based on well-known ideas and its benefits are completely not obvious.

I guess that the paper can be published, but only after issues in a) are addressed.

**Experience Assessment:**

I have published one or two papers in this area.

**Review Assessment: Checking Correctness Of Derivations And Theory:**

I assessed the sensibility of the derivations and theory.

**Review Assessment: Checking Correctness Of Experiments:**

I assessed the sensibility of the experiments.

**Review Assessment: Thoroughness In Paper Reading:**

I read the paper at least twice and used my best judgement in assessing the paper.

---

> ### Author Response · Authors · 2019-11-15
> **Response reviewer 4: Part 1**
>
> We would like to thank the reviewer for the review and helpful suggestions.
>
> 1) “From the description of the functionals L1 and L2 (bottom of page 4 and top of the page 5) the reader can think that the authors tune parameters (zeta_1,theta_1) and (zeta_2,theta_2) for each sample point separately”
>
> We thank the reviewer for pointing this out, this is a typo and we changed this in the paper.
>
> 2) “The authors claimed that the experiments were done both for kernel features and for explicit features based on neural networks. However, in the experimental section there are no results obtained when using implicit kernels. Nothings is told on how to select kernel parameters.”
>
> Experiments with implicit feature maps where already shown on Figure 4a. Table 1 in the appendix shows the hyperparameters used for the experiments. The bandwidth of the Gaussian kernel for generation corresponds to the bandwidth that gave the best performance determined by cross-validation on the MNIST classification problem.
>
> 3) “The authors claimed that thanks to PCA-like definition of latent vectors they are orthogonal which is similar to disentangle representations. However, there are no any empirical evidence whether it is possible to benefit somehow from that orthogonal property, as well as there is no comparison with approaches to construct disentangleв latent representation for other types of generative models.”
>
> The definition of disentanglement in the literature is not that precise. However many believe that a representation with statistically independent variables is a good starting point [1,2]. This already gives an indication that the model could resemble a disentangled representation. This is confirmed by the empirical evidence on Figure 5. We explore the learned uncorrelated-features by traversing along the (orthogonal) eigenvectors on the celebA and Dsprites dataset. These are common datasets to demonstrate disentanglement, where we repeat the experiments of [3] for the proposed Gen-RKM. For the Dsprites dataset, notice that the first and second components correspond to the y and x positions respectively. Rows 3 and 4 show the same for hearts. On the celebA dataset, rows 5 and 6 shows the reconstructed images while traversing along the principal components. When moving along the first component from left-to-right, the hair-color of the woman changes, while preserving the face structure. Whereas traversal along the second component, transforms a man to woman while preserving the orientation. When the number of principal components were 2 while training, the brightness and background light-source corresponds to the two largest variances in the dataset. This small example demonstrates how we can interpret the different components of the latent representation learned by the Gen-RKM. The latent space dimension in the RKM setting has a similar interpretation as the number of hidden units in a restricted Boltzmann machine, where in the specific case of the RKM these hidden units are uncorrelated.

---

> ### Author Response · Authors · 2019-11-15
> **Response reviewer 4: Part 2**
>
>
> 4) “The main idea of the paper.”
>
> A novel multi-view generative model based on the RKM framework where multiple views of the data can be generated simultaneously. The model incorporates the best of both worlds i.e. kernel methods and neural networks (e.g. convolutional layers). This is done based on the RKM framework. The model gives us uncorrelated latent variables, which gives an indication that the Gen-RKM could resemble a disentangled representation. The example on Figure 5 supports the latter.
>
> There is also an intuitive connection between Gen-RKM and autoencoders.
> Namely, the  properties of kernel PCA resemble the objective of the 3 variations of an autoencoder: standard [4], VAE [6] and beta-VAE [7].
> 4.1) Similar to an autoencoder, when using an explicit feature map Gen-RKM tries to minimize the reconstruction error in the loss function (see Eq. 9), where kernel PCA acts as a denoiser (the information is compressed in the first components).
> 4.2) By interpreting kernel PCA in the LS-SVM setting, the kernel PCA analysis can take the interpretation of a one-class modeling problem with zero target value around which one maximizes the variance [5]. When choosing a good feature map, one thus expects the latent variables to be Gaussian distributed around zero (We also observed this in the experiments). This resembles the added regularization term in the objective of the VAE [6],  which is expressed as the Kullback-Leibler divergence between the returned distribution and a standard Gaussian, i.e. we want the latent variables of the VAE to be Gaussian distributed.
> 4.3) Kernel PCA gives us uncorrelated components in feature space. While it was already shown that linear PCA does not give a good disentangled representation for images [7,8], we feel that this is not a fair comparison as a linear kernel is not appropriate for image data. By designing a good kernel (e.g. Convolutional layers) and doing kernel PCA with an explicit feature map, it is possible to get a disentangled representation for images as we show on the example in Figure 5.
>
> By resembling the components of the 3 variations of the autoencoder, the
> Gen-RKM performs comparably on the 3 objectives simultaneously :
> reconstruction, generation and disentanglement. We added a small paragraph
> to highlight these differences.
>
> 5) “The authors did not provide any quantitative measure and comparison with existing approaches;”
>
> We thank you for the suggestion and agree with the comment. We added the
> following experiments:
> 5.1) To assess the quality of generation, we compare Gen-RKM with standard VAE [6] on the MNIST and celebA dataset. To have a fair comparison, the models have the same encoder/decoder, optimization parameters and the same loss function: binary cross-entropy. Both models were trained until convergence. The performance is evaluated qualitatively by comparing reconstruction and random sampling (similar to the experiments in [10]). Quantitatively, we compare the models using the Fréchet Inception Distance (FID) [9], which is a common metric to evaluate quality of generation. We observed empirically that FID scores are better for the Gen-RKM when using the same encoder/decoder architecture and small latent dimension. This is confirmed by a qualitatively evaluation, where the VAE generates more smoothed images.
> 5.2) To assess the disentanglement, we compare Gen-RKM with VAE [6] and beta-VAE [8] on the Dsprites [11] and Teapot dataset [7]. Again, the 3 models have the same size of the encoder and decoder architectures, same number of latent variables,.... The  performance is measured by the proposed framework of Eastwood et al. [7], which gives 3 measures: disentanglement, completeness and informativeness. We observe empirically that the Gen-RKM has good performance when the latent space dimension is well chosen.
>
> We hope this addresses the reviewer’s concerns.

---

### Author Response · Authors · 2019-11-15
**Response to all reviewers**

We would like to thank the reviewers for their review and helpful suggestions.

Based on the comments, we made the following additions to the paper:
1) An intuitive connection between Gen-RKM and autoencoders.

2) To assess the quality of generation, we compared Gen-RKM with standard VAE [6] on the MNIST and celebA dataset. The comparison is done qualitatively, along with a quantitative comparison using the Fréchet Inception Distance (FID) [9]. We observed empirically that the FID scores are better for the Gen-RKM when using the same encoder/decoder architecture and small latent space dimension. This is confirmed by the qualitatively evaluation, where the VAE generates smoother images.

3) To assess the disentanglement, we compared Gen-RKM with VAE [6] and beta-VAE [8] on the Dsprites [11] and Teapot dataset [7]. The performance is measured by the proposed framework of Eastwood et al. [7], which gives 3 measures: Disentanglement, Completeness and Informativeness. We observe empirically that the Gen-RKM has good performance when the latent space dimension is well chosen.

We hope this addresses the reviewers’ concerns.

References used in all responses:
[1] Schmidhuber, Jürgen. "Learning factorial codes by predictability minimization." Neural Computation 4.6 (1992): 863-879.
[2] Karl Ridgeway. A survey of inductive biases for factorial representation-learning. CoRR, abs/1612.05299, 2016.
[3] Higgins, Irina, et al. beta-VAE: Learning Basic Visual Concepts with a Constrained Variational Framework. ICLR, 2017, 2.5: 6.
[4] Kramer, Mark A. "Nonlinear principal component analysis using autoassociative neural networks." AIChE journal 37.2 (1991): 233-243.
[5] Suykens, Johan AK, et al. "A support vector machine formulation to PCA analysis and its kernel version." IEEE Transactions on neural networks 14.2 (2003): 447-450.
[6] Diederik P. Kingma and Max Welling. Auto-Encoding Variational Bayes. In 2nd International Conference on Learning Representations, ICLR 2014, Banff, AB, Canada, April 14-16, 2014.
[7] Eastwood, Cian, and Christopher KI Williams. "A framework for the quantitative evaluation of disentangled representations." (2018).
[8] Higgins, Irina, et al. "beta-VAE: Learning Basic Visual Concepts with a Constrained Variational Framework." ICLR 2.5 (2017): 6.
[9] Heusel, Martin, et al. "GANs trained by a two time-scale update rule converge to a local Nash equilibrium." Advances in Neural Information Processing Systems. 2017.
[10] Tolstikhin, Ilya, et al. "Wasserstein auto-encoders." arXiv preprint arXiv:1711.01558 (2017).
[11] Matthey, Loic, et al. "Dsprites: Disentanglement testing sprites dataset." URL https://github. com/deepmind/dsprites-dataset/.[Accessed on: 2018-05-08] (2017).

---

### Decision · Program_Chairs · 2019-12-19

**Decision:**

Reject

**Comment:**

The paper proposes a way to use kernel method for multi-view generation. The points are mapped into a common subspace (with CNN feature extractor and kernel on top), and then a generation procedure from a latent point is given.
I found the paper not easy to ready and follow; the idea of using CNN + kernel methods have been around for some years (for example, see "Impostor networks" by Lebedev et. al), and explicit feature map shows that kernel is just an additional layer to the network. Overall, the approach is straightforward, the generation can be quite slow and the benefits are not clear. The reviewers are mildly negative, so I think this time this paper can not be accepted.

---

> ### Author Response · Authors · 2020-02-13
> **Author Response**
>
> We thank the meta-reviewer for the comments. Here we would like to express our concerns with the statements in the decision comment:
>
> 1). We propose to use kernel methods for multi-view generation ‘and’ uncorrelated feature learning.
>
> 2). The points are not mapped into a common subspace. Rather, they are mapped to the feature-spaces and the eigendecomposition of the concatenated kernel matrices represents the common subspace.
>
> 3). Moreover, all 3 reviewers agree that the ‘paper is easy to read’.
>
> 4). The reference “Imposter Networks” is not particularly relevant in this regard (it focuses on classification tasks for low-power devices). As opposed to Imposter Networks, the kernel in our model is not an additional layer to the network, rather it is inherent to the model.
>
> 5). Generation is very fast when using explicit feature-maps. In this case, it just means decoding the latent-vectors (by passing through the transposed-CNN) and it scales linearly with the desired number of generated samples. However, when using implicit feature-maps, pre-image problem needs to be solved and depending on the method chosen, the generation time would vary.
>
> 6). Benefits include new synergy between (deep) neural networks and kernel methods, modelling a common subspace from different data sources enabling multi-view generation and disentangled (uncorrelated due to PCA in feature space) feature learning, all within the same model. These are validated through various experiments (qualitative & quantitative).
>
> 7). Lastly, Rev #4 upgraded the score based on our updates, Rev #2 was the most positive but never responded post-rebuttal (the last response was Nov 5) and Rev #1 was asking for classification performance, which is not in the scope of this current work (unsupervised learning/generative modelling).